# Direct Preference Density Alignment for Conversational Audio Equalization

## Abstract

Large Language Model alignment methods typically rely on learned proxy reward models, which are often unstable and fail to capture the multimodal subjectivity inherent in continuous-control tasks. We propose an alternative that removes the proxy entirely, aligning models through preference density maps constructed directly from large-scale user data ($N \approx 90,000$), and instantiate this approach on a continuous audio equalization control task. Through comparative analysis, we demonstrate that standard offline Direct Preference Optimization (DPO), in contrast to online Group Relative Policy Optimization (GRPO), struggles to adhere to pre-defined output formats. Building on this insight, we introduce a hybrid GRPO+DPO training pipeline that combines the proactiveness of online optimization, with the targeted refinement of offline methods. Our framework achieves the best performance across objective metrics and, in a blind A/B listening test, enables a 1.5B-parameter model (Qwen 2.5) to achieve comparable performance to a carefully prompt-engineered GPT-4o mini baseline.

## 1. Introduction

The integration of Large Language Models (LLMs) into continuous control systems bridges the "semantic gap" between high-level human intent and low-level parameter execution. In subjective domains like audio reproduction, where abstract descriptors (e.g., "warm", "bright") are mapped to technical parameters, a single prompt rarely corresponds to a deterministic point. Instead, it represents a distribution of valid user preferences (Cartwright & Pardo, 2013).

However, aligning LLMs to these subjective distributions remains an open challenge. Modern approaches such as Reinforcement Learning from Human Feedback (RLHF)

(Ouyang et al., 2022) rely on learning a proxy Reward Model (RM). In continuous control tasks, these proxies are prone to "reward hacking," where the learned policy exploits inaccuracies in the reward approximation rather than learning the true utility (Gao et al., 2023). Although Direct Preference Optimization (DPO) (Rafailov et al., 2023) attempts to bypass the need for an explicit reward model, we find that it often struggles to maintain structural constraints when operating in continuous, real-valued coordinate spaces without a validity penalty to guide the optimization.

In this work, we propose a framework that eliminates the learned proxy entirely. Instead, we leverage population data to construct preference density maps using Reflective Kernel Density Estimation (Reflective-KDE). This approach transforms raw preference data into a non-parametric, ground-truth reward surface that inherently respects the underlying control boundaries.

Using this reward surface, we conduct a rigorous evaluation of alignment strategies. In particular, we compare the stability of offline DPO with the online, group-based exploration of Group Relative Policy Optimization (GRPO) (Shao et al., 2024). Our empirical results reveal a critical dichotomy: DPO can contribute to preference sharpening but struggles with format adherence, whereas online group methods (GRPO) maintain superior structural adherence through self-correction but are prone to globally mediocre "safe" predictions. To reconcile these limitations, we introduce a hybrid GRPO+DPO pipeline, demonstrating that initializing with the structural robustness provided by GRPO, followed by preference refinement through DPO, achieves peak performance.

Our contributions are summarized as follows:

- **Non-Parametric Reward Framework:** We introduce a method for aligning LLMs in continuous spaces using Reflective-KDE, bypassing the instability of learned Reward Models.

- **Evaluation of Online vs. Offline Alignment:** We identify specific failure modes of offline DPO in continuous control and demonstrate the structural superiority of online group methods.

- **Hybrid Implementation & Perceptual Validation:**

---
[1]Anonymous Institution, Anonymous City, Anonymous Region, Anonymous Country. Correspondence to: Anonymous Author <anon.email@domain.com>.

Preliminary work. Under review by the International Conference on Machine Learning (ICML). Do not distribute.

We validate a hybrid GRPO+DPO pipeline via a blind A/B listening test, showing that a 1.5B parameter model can achieve perceptual parity with GPT-4o mini.

## 2. Related Work

Our work positions itself at the intersection of semantic audio control, preference alignment, and reinforcement learning in continuous action spaces.

### 2.1. From Deterministic Regression to Distributional Control

Traditionally, mapping natural language to audio parameters has been treated as a supervised regression problem. Early systems relied on fixed vocabulary-to-parameter mappings (Pardo et al., 2012), while recent approaches employ LLMs to interpret unconstrained text and predict point-estimates for equalization (EQ) and reverberation (reverb) parameters (Chu et al., 2025; Doh et al., 2025). Crucially, these regression-based methods implicitly favor a single parameter realization at inference time. Prior work (Author, 2026) challenged this assumption, proposing that audio descriptors like "warm" map to distributions rather than points. However, that study relied on small-scale supervised learning ($N = 11$). Our work scales this distributional premise ($N \approx 3,000$ per prompt) and transitions from supervised matching to direct Reinforcement Learning (RL), enabling on-policy optimization that fully exploits densely annotated data.

### 2.2. Proxy-Free and Direct Alignment

The standard LLM RL-tuning pipeline (Ouyang et al., 2022) typically relies on Proximal Policy Optimization (PPO) (Schulman et al., 2017) optimized against a learned Reward Model (RM). However, PPO requires training a separate Value Network to estimate the expected return. In subjective, multimodal domains like audio preference, fitting a Value Network introduces an additional proxy objective, compounding the risk of reward misspecification in subjective domains (Casper et al., 2023). To circumvent the instability of critics and reward models, recent work has shifted toward direct alignment methods. Approaches like DPO, Identity Preference Optimization (IPO) (Azar et al., 2023), and Kahneman-Tversky Optimization (KTO) (Ethayarajh et al., 2024) optimize the policy directly from preference pairs without an explicit reward function. However, these methods are blind to the probability mass assigned to completions outside the preference dataset. As a result, the model may increase the probability assigned to malformed or constraint-violating outputs during training, even though such outputs are never explicitly preferred. As we demonstrate, this lack of active exploration makes them brittle in continuous control tasks, where the model must actively

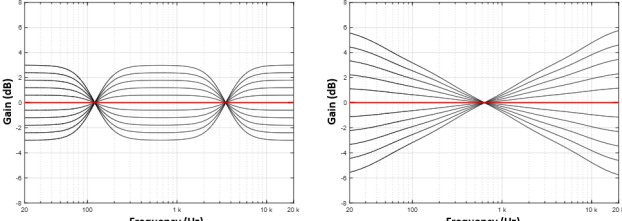

*Figure 1.* The filters that make up the EQ controller. Horizontal movement in the $[-6, 6]^2$ plane produces a "smile curve" effect (left), whereas vertical movement applies a linear adjustment (right).

learn to maintain strict formatting constraints.

### 2.3. Group Relative Policy Optimization

GRPO (Shao et al., 2024) offers a middle ground between the heavy machinery of PPO and the static nature of DPO. Originally designed for reasoning tasks, GRPO eliminates the PPO value network entirely. Instead of a learned critic, it uses the average reward of a group of generated rollouts as the baseline. We find this approach particularly well-suited for our framework, which attempts to model subjective audio control. By generating its own data during training, GRPO provides the exploration necessary to learn format constraints.

## 3. Preliminaries and Reward Surface Construction

To effectively align an LLM with subjective auditory preferences, we must first define the control space and construct a reliable reward signal that captures the nuance of human perception without relying on unstable proxy networks.

### 3.1. The Beosonic Control Space

We ground our study in the "Beosonic" interface (Bang&Olufsen, 2023), a continuous two-dimensional control plane $\mathcal{X} \in [-6, 6]^2$. This space reduces the EQ transfer function to 2 parameters, with each corresponding to the intensity of a distinct equalization filter ($\pm 6$ gain), as illustrated in Figure 1. With this approach we effectively represent audio equalization through an interpretable 2D parameter space. Unlike discrete selection tasks, the model must predict a continuous coordinate $x \in [-6, 6]^2$.

### 3.2. Dataset and Motivation

Our reward signal is grounded in a large-scale comparison dataset, originally collected to evaluate the framework proposed in (Author, 2026). In this deployment, participants were presented with natural language prompts paired

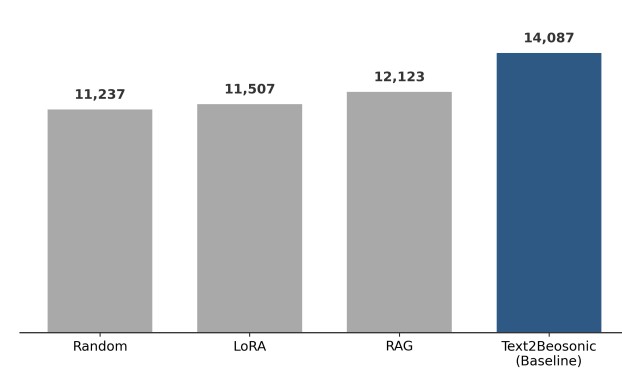

*Figure 2.* Total user selection counts by model type. The Zero-Shot baseline (T2B) outperforms complex distributional models (LoRA, RAG), indicating that previous attempts at distributional modeling failed to capture high-utility preference modes due to data scarcity.

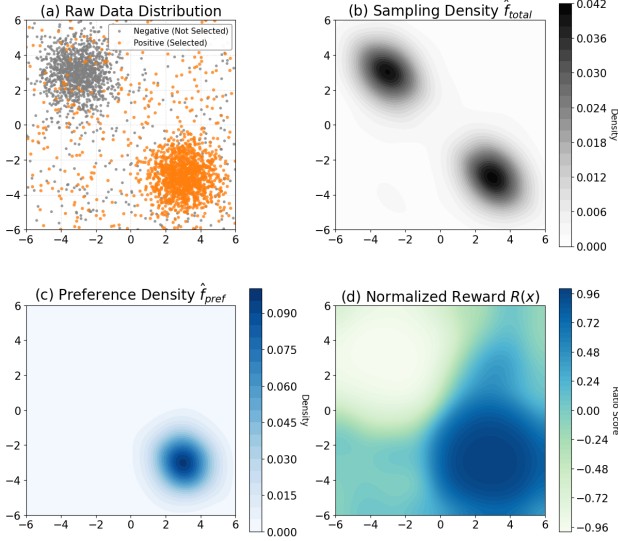

*Figure 3.* **Visualization of the Non-Parametric Reward construction on simulated data.** **(a)** Raw observations illustrate a scenario with significant sampling bias: the Top-Left mode is heavily explored by agents but rarely selected by users (Negative/Gray), while the Bottom-Right mode is both heavily explored and frequently selected (Positive/Orange). **(b)** Density estimation on the full dataset ($\hat{f}_{total}$) provides an "image" of which locations are "visited" more frequently. **(c)** The density of positive samples ($\hat{f}_{pref}$). As illustrated, estimating only on the positive examples fails to capture information on the dispreferred upper-left corner. **(d)** The final Reward Surface derived via Probability of Selection ($S(x)$). By normalizing the preference density against the sampling density, the method successfully cancels out the visitation bias. The Top-Left "false positive" is suppressed, and the reward signal is correctly concentrated on the true high-utility mode in the Bottom-Right.

with audio variations generated by four distinct strategies: a deterministic Zero-Shot LLM (T2B), two distributional baselines (LoRA, RAG), and a Random agent. For each variation, users provided binary feedback indicating whether the audio satisfied the prompt.

The resulting dataset comprises $N \approx 90,000$ interaction events (consisting of both selections and rejections). An analysis of the positive selections (Figure 2) reveals a critical insight: The prompt-engineered Zero-Shot baseline (T2B) achieved the highest selection count ($14,087$), significantly outperforming the more complex distributional models. This outcome suggests that the prior distributional approaches, having been trained on extremely sparse data ($N = 11$), failed to generalize effectively.

This observation motivates our RL approach: We require an objective that can explicitly optimize for the peaks of human satisfaction (Exploitation) while maintaining sufficient coverage of the space (Exploration).

### 3.3. Probability of Selection

To construct the reward function $R(x|p)$ for a specific prompt $p$, we cannot simply estimate the density of high-rated points, as this would be biased by the sampling distribution of the generating models (e.g., if a model samples the center frequently, high density there might differ from high preference).

We therefore aim to estimate the Probability of Selection $P(\text{Selected}|x)$ for any given coordinate x. Let $\mathcal{D}_{all}$ be the set of all offered coordinates for prompt $p$, and $\mathcal{D}_{pref} \subseteq \mathcal{D}_{all}$ be the same set weighted by binary selection events. We estimate two densities using Reflective-KDE (defined in Subsection 3.4)

1. $\hat{f}_{total}(x)$: The density of the proposal distribution

(where the points came from; $P(x)$).

2. $\hat{f}_{pref}(x)$: The density of the users' preferred regions ($P(x|\text{Selected})$).

By applying Bayes' theorem, the Probability of Selection is proportional to the ratio of the positive density to the total offered density:

$$\begin{aligned} S(x) &= P(\text{Selected} \mid x) \\ &= \frac{P(x \mid \text{Selected})\, P(\text{Selected})}{P(x)} \\ &\propto \frac{\hat{f}_{\text{pref}}(x)}{\hat{f}_{\text{total}}(x) + \epsilon} \end{aligned} \tag{1}$$

where $\epsilon$ ensures numerical stability. This ratio effectively cancels out the sampling bias, isolating the underlying population preference.

### 3.4. Reflective Kernel Density Estimation

Standard KDE suffers from boundary bias in bounded spaces, underestimating density near the edges (where user preferences often cluster, e.g., "Max Bass"). To correct this, we implement Reflective-KDE (Schuster, 1985). We augment the dataset by reflecting all points $x_i$ across the boundaries of $\mathcal{X}$ into the 8 surrounding neighbor squares, creating a $3 \times 3$ grid of "ghost" points. The density estimate becomes:

$$\hat{f}(x) = \frac{1}{9nh^2} \sum_{i=1}^{n} \sum_{k=0}^{8} K\left(\frac{x - T_k(x_i)}{h}\right) \qquad (2)$$

where $T_k$ represents the reflection mappings ($T_0$ is the Identity) and $K$ is a Gaussian kernel. We set the bandwidth $h = 0.25 \cdot h_{\text{Scott}}$. Standard estimators like Scott's Rule (Scott, 2010) assume unimodal Gaussianity, which results in excessive smoothing when applied to the highly multi-modal landscape of subjective preference. By scaling $h$, we trade asymptotic density accuracy for topological fidelity, ensuring that distinct local optima are preserved rather than merged into a single centroid.

### 3.5. Global Normalization and Augmentation

To ensure the reward signal is consistent across different prompts, we perform a global normalization pass. We compute the global maximum ($S_{max}$) and minimum ($S_{min}$) density ratios across the entire dataset. The final reward $R(x)$ used for RL training is scaled to $[-1, 1]$:

$$R(x) = 2 \cdot \frac{S(x) - S_{min}}{S_{max} - S_{min}} - 1 \qquad (3)$$

The whole construction process is visualized in Figure 3.

Finally, we employ semantic data augmentation. We assume that synonymous prompts (e.g., "Make it warmer" and "Increase warmth") share an identical preference manifold. We aggregate data for synonymous clusters and map them to the same reward surface.

### 3.6. Evaluation Splits

To assess generalization, we partition the prompts into three sets; a training set containing approximately $80\%$ of the prompts used for policy updates, a validation set consisting of the randomly held-out prompts drawn from the same distribution, and a manually isolated Out-Of-Distribution (OOD) set containing a cluster of prompts related to "Vocal Clarity." These OOD concepts are entirely unseen during training, allowing to evaluate the model's ability to extrapolate its learned preference geometry to novel semantic inputs.

## 4. Alignment Strategies in Continuous Spaces

Having constructed a non-parametric reward surface $R(x)$, our goal is to align an LLM policy $\pi_\theta$ to maximize the expected return $\mathbb{E}_{x \sim \pi_\theta}[R(x)]$. In this section, we analyze the structural limitations of standard offline alignment in continuous spaces and propose a hybrid pipeline that utilizes an online-aligned model to synthesize its own high-precision preference dataset.

### 4.1. The Challenge of Continuous Formatting

Unlike standard chat models, our agent must output valid coordinates in a strict format (e.g., `[1.5, -3.2]`). A critical failure mode in this domain is *format collapse*, where the model drifts into generating conversational text or invalid syntax.

We hypothesize that offline methods (i.e. DPO) are particularly susceptible to this. Since DPO is limited to the specific negative examples in the static dataset, it cannot penalize emergent structural failures. As the policy shifts, it may drift into generating novel malformed syntaxes that were not present in the pre-collected data, effectively bypassing the optimization constraints. Conversely, online methods (like GRPO) generate their own predictions during training. This allows the environment to provide immediate negative feedback ($R = -1.0$) for syntax errors, effectively "grounding" the model in the constraints of the control interface.

### 4.2. Baseline 1: Direct Preference Optimization

DPO optimizes the policy by increasing the margin between preferred responses $y_w$ and dis-preferred ones $y_l$:

$$\mathcal{L}_{\text{DPO}} = -\mathbb{E}_{(x,y_w,y_l) \sim \mathcal{D}} \left[\log \sigma\left(\hat{r}_\theta(x, y_w) - \hat{r}_\theta(x, y_l)\right)\right] \qquad (4)$$

Where the implicit reward is defined as:

$$\hat{r}_\theta(x, y) = \beta \log \frac{\pi_\theta(y|x)}{\pi_{\text{ref}}(y|x)}.$$

While efficient, DPO relies heavily on the Kullback-Leibler (KL)-divergence constraint (implicitly controlled by $\beta$) to prevent the model from deviating from the reference policy $\pi_{\text{ref}}$. In our domain, we define the dataset $\mathcal{D}$ using the reward map $R(x)$, where $y_w$ and $y_l$ are pairs of coordinates such that $R(y_w|p) > R(y_l|p)$. More details are provided in Subsection 4.4.

### 4.3. Baseline 2: Group Relative Policy Optimization

We adopt GRPO as our online framework. Unlike PPO, GRPO eliminates the value model. For each prompt $q$, the policy generates a group of $G$ outputs $\{o_1, \ldots, o_G\}$. By setting $\rho_i(\theta) = \frac{\pi_\theta(o_i|q)}{\pi_{\text{old}}(o_i|q)}$

The objective maximizes:

$$\mathcal{J}_{\text{GRPO}}(\theta) = \mathbb{E}_{q \sim P(Q), \{o_i\}_{i=1}^G \sim \pi_{\theta_{\text{old}}}}$$

$$\left[ \frac{1}{G} \sum_{i=1}^{G} \left( \underbrace{\min\left(\rho_i(\theta)A_i, \text{clip}(\rho_i(\theta), 1-\epsilon, 1+\epsilon)A_i\right)}_{\text{Clipped Surrogate Objective}} \right. \right.$$

$$\left. \left. - \underbrace{\beta \mathbb{D}_{\text{KL}}(\pi_\theta || \pi_{\text{ref}})}_{\text{KL Penalty}} \right) \right] \tag{5}$$

The advantage $A_i$ is computed by normalizing the rewards within the group. Crucially, if the model generates an invalid format, the reward function returns a penalty ($R = -1.0$). Since this penalty is included in the group normalization, the model explicitly learns to avoid invalid syntax to achieve a positive advantage relative to its peers.

### 4.4. Proposed Method: Hybrid GRPO+DPO Pipeline

We observe that while GRPO excels at structural stability (learning *how* to format), it may fail to find all the local maxima of the preference density (learning exactly *what* different people want to hear). To resolve this, we propose a two-stage pipeline that combines the stability of online RL with the precision of offline RL.

**Stage 1: Structural Alignment (GRPO).** We first train $\pi_\theta$ using GRPO against the Density Map. This transforms the generalist LLM into a robust control agent ($\pi_{GRPO}$) that reliably outputs coordinates in the correct manifold.

**Stage 2: Synthetic Preference Mining & Peak Injection.** We construct a synthetic DPO dataset $\mathcal{D}_{syn}$ using $\pi_{GRPO}$ as the generator. To ensure the dataset steers the model toward the local optima, we employ a peak injection strategy:

1. **On-Policy Mining:** We sample completions from $\pi_{GRPO}$. Those with high rewards ($R(x) > 0.8$) are added to the positive pool; malformed or low-reward outputs ($R(x) < -0.2$) are added to the negative pool.

2. **Peak Injection:** If $\pi_{GRPO}$ fails to discover the top-K local maxima of the Density Map $R(x)$ for a given prompt, we analytically find the peak coordinates via grid search on $R(x)$ and inject them into the positive pool as "Golden Truths."

The process is exactly the same as in standard DPO (Subsection 4.2), with $\pi_{ref}$ as the generator instead of $\pi_{GRPO}$.

**Stage 3: Refinement (DPO).** We fine-tune the model using DPO on $\mathcal{D}_{syn}$, setting $\pi_{ref} = \pi_{GRPO}$. By initializing with the structurally robust GRPO model and training on injected expert peaks with relatively high $\beta$ values, the hybrid

model achieves both high format compliance and improved preference matching.

## 5. Experiments and Results

We evaluate our alignment strategies on the OOD test set described in Section 3.6. This set contains semantic concepts unseen during training, requiring the model to generalize the geometric understanding of the preference space.

### 5.1. Experimental Setup

**Models:** We utilize the **Qwen2.5-Instruct** family (0.5B and 1.5B parameters) (Team, 2024) as our base models due to their strong instruction-following capabilities and open weights.

**Baselines:** We compare against In-Context Learning (ICL) baselines using Qwen2.5 and GPT-4o mini. To establish a rigorous performance floor, we inject domain knowledge into these baselines via a sophisticated system prompt containing an explicit semantic-to-coordinate mapping dictionary (e.g., "warm" $\rightarrow [0, -6]$), derived from audio attribute clustering literature (Francombe et al., 2017). In contrast, our RL-tuned models are agnostic to these expert definitions; they rely on a minimal system prompt and must learn the semantic-to-acoustic relationships purely from the reward signal.

**Metrics:**

**Greedy Mean Relative Reward (GMRR):** We evaluate the models' performance on their most probable (**greedy**) completion for each prompt. Additionally, while training utilizes globally normalized rewards, evaluation requires measuring optimality **relative** to the specific prompt. We therefore revert to local normalization: For each prompt, the reward is rescaled such that the maximum of the density map is $1.0$ and the minimum is $0.0$.

$$\text{GMRR}(x) = \frac{S(x) - S_{min,local}}{S_{max,local} - S_{min,local}}. \tag{6}$$

Under this metric, a score of $1.0$ indicates the model found the absolute peak of human preference for that specific query, while the absolute low or malformed outputs are assigned $0.0$.

**Format Success Rate (Parse Rate):** The percentage of generated outputs that **strictly** adhere to the coordinate format `[x, y]`.

### 5.2. The Stability Gap: Online vs. Offline

Our first experiment investigates the structural stability of the alignment algorithms. We sweep the KL-divergence penalty coefficient $\beta \in \{0.0, 0.1, 0.5, 1.0\}$ for both DPO

*Table 1.* Greedy Mean Relative Reward ($\pm$ Standard Deviation) on the OOD test set and format success rate (%). The top section compares the GRPO and DPO methods for different $\beta$ values. The bottom section provides ICL models reference scores.

| $\beta$ | Model | 12-GRPO | | DPO | |
|---|---|---|---|---|---|
| | | GMRR | Parse Rate | GMRR | Parse Rate |
| 0.0 | Qwen2.5-0.5B | $0.44 \pm 0.08$ | 100% | 0.00 | 0% |
| | Qwen2.5-1.5B | $0.52 \pm 0.10$ | 100% | 0.00 | 0% |
| 0.1 | Qwen2.5-0.5B | $0.49 \pm 0.14$ | 100% | $0.31 \pm 0.27$ | 47% |
| | Qwen2.5-1.5B | $0.53 \pm 0.09$ | 100% | 0.00 | 0% |
| 0.5 | Qwen2.5-0.5B | $\mathbf{0.52} \pm 0.10$ | 100% | $0.33 \pm 0.22$ | 81% |
| | Qwen2.5-1.5B | $\mathbf{0.53} \pm 0.09$ | 100% | $0.32 \pm 0.27$ | 69% |
| 1.0 | Qwen2.5-0.5B | $0.50 \pm 0.10$ | 94% | $0.30 \pm 0.25$ | 98% |
| | Qwen2.5-1.5B | $0.51 \pm 0.09$ | 100% | $0.07 \pm 0.23$ | 10% |
| **ICL Reference Models** | | **GMRR** | | **Parse Rate** | |
| | Qwen2.5-0.5B | $0.25 \pm 0.24$ | | 54% | |
| | Qwen2.5-1.5B | $0.49 \pm 0.08$ | | 100% | |
| | GPT-4o mini | $\mathbf{0.56} \pm 0.07$ | | 100% | |

(Offline) and GRPO (Online) and measure the Format Success Rate on the OOD test set.

As shown in Table 1, a critical failure mode emerges for DPO. A low $\beta$ value (0.1 and 0.0) allows the model to deviate more aggressively from its initial state. During training, the objective of the algorithm is to maximize the margin between the positive and negative examples in the pre-developed preference dataset. This approach indeed increases the relative probability of generating a preferred example, *but only compared to generating a dis-preferred one*. It is likely, especially with low KL regularization, that the overall probability of generating a positive example decreases during training. However, the training objective is still met as long as the overall probability of the negative counterpart decreases even faster.

A different way to view this is that offline DPO "tunnel-visions" on the two examples in the preference dataset entry, while the probabilities of all other completions are updated without any supervision.

In contrast, GRPO maintains 100% format success rate even at $\beta = 0.0$. Because the online agent generates its own data during training, any format deviation results in an immediate penalty ($R = -1$), forcing the policy to remain within the valid syntax manifold. Interestingly, a "too high" $\beta$ value can restrain the model too much and won't allow strict structure following.

### 5.3. Scaling Exploration

We further analyze the impact of the group size $G$ (number of rollouts per prompt) on GRPO performance (Table 2). Increasing $G$ consistently improves the GMRR, saturating between $G = 8$ and $G = 16$. However, this improvement is relatively small, with even the 2-Rollout models outperforming their sophisticated ICL counterparts. This observation

aligns with recent studies (Wu et al., 2025), which question the benefits of larger samples sizes relative to the computational burden.

### 5.4. Hybrid Pipeline Performance

Finally, we evaluate the proposed **hybrid GRPO+DPO** pipeline. We select the best GRPO configuration ($\beta = 0.5$) as the initialization and RL-tune it using DPO ($\beta = 1.0$, post-GRPO as Reference Policy) on the synthetically mined "Peak Injection" dataset (Section 4) across different rollouts.

Results in Table 3 demonstrate that the hybrid approach achieves a new highest reward. For the 0.5B model, GMRR jumps from 0.53 (Base GRPO) to **0.56** (hybrid), matching the GPT-4o baseline (0.56) but at the cost of some inconsistency (97% Parse Rate). However, the 1.5B model achieves a GMRR of **0.60** without compromising strict format compliance (100% Parse Rate). This validates our core hypothesis: GRPO provides the necessary structural foundation, enabling offline DPO to sharpen preference alignment.

To assess if the preference manifold could be learned via supervision, we also trained a Supervised Fine-Tuned (SFT) baseline with Low Rank Adaptation (Hu et al., 2021) (LoRA) on the "peaks" of the training prompts (the same as DPO). This model achieved a GMRR of 0.45, performing worse than the base model with In-Context Learning (0.49).This suggests that SFT leads to overfitting on specific coordinate clusters, degrading the model's ability to extrapolate to OOD semantic concepts. Table 4 provides a performance summary of the different methods for Qwen2.5-1.5B.

## 6. Subjective Evaluation

While our objective metrics demonstrate the convergence of the proposed RL approach, the ultimate validation of an audio system lies in listener perception. We conducted a blind A/B listening test to compare our best-performing hybrid model (Qwen2.5-1.5B) against the baseline (GPT-4o mini via In-Context Learning). Both models controlled the exact same equalization system, ensuring that any perceived differences result purely from the parameters selected by the LLMs.

### 6.1. Unconstrained Data Collection and Protocol

To ensure the evaluation reflected real-world interaction, we utilized a two-phase study with human listeners. First, we recruited 12 participants for a prompt-elicitation phase. We employed an unconstrained elicitation protocol; participants listened to presented audio clips and were asked to request *any* changes they desired without being informed of the system's technical limitations (i.e., that it was an equalizer).

*Table 2.* Performance comparison of Qwen2.5 models under the GRPO algorithm with varying numbers of rollouts (sample size) and fixed $\beta$ value (0.5). Metrics shown are Greedy Mean Relative Reward ($\pm$ Standard Deviation) and format success rate (%).

| | 2 Rollouts | | 4 Rollouts | | 8 Rollouts | | 16 Rollouts | | 32 Rollouts | |
|---|---|---|---|---|---|---|---|---|---|---|
| Model | GMRR | Parse Rate | GMRR | Parse Rate | GMRR | Parse Rate | GMRR | Parse Rate | GMRR | Parse Rate |
| Qwen2.5-0.5B | $0.47 \pm 0.17$ | 81% | $0.50 \pm 0.14$ | 93% | $\mathbf{0.53} \pm \mathbf{0.10}$ | 100% | $0.51 \pm 0.10$ | 100% | $0.48 \pm 0.14$ | 96% |
| Qwen2.5-1.5B | $0.50 \pm 0.08$ | 100% | $0.51 \pm 0.09$ | 100% | $0.51 \pm 0.09$ | 100% | $\mathbf{0.52} \pm \mathbf{0.09}$ | 100% | $0.52 \pm 0.09$ | 100% |

*Table 3.* Performance comparison of Qwen2.5 models using GRPO (Baseline) versus the combined GRPO + DPO algorithm. The $\beta$ values for GRPO and DPO are fixed at 0.5 and 1.0 respectively.

| Model | Rollouts | Approach | GMRR | Parse Rate (%) | Observation |
|---|---|---|---|---|---|
| Qwen2.5-0.5B | 2 | GRPO | $\mathbf{0.47} \pm 0.17$ | 81 | |
| | | **GRPO+DPO** | $0.44 \pm 0.32$ | **88** | Drop in GMRR, improvement in parsing rate |
| | 4 | GRPO | $0.50 \pm 0.14$ | **93** | |
| | | **GRPO+DPO** | $\mathbf{0.56} \pm 0.26$ | 73 | Preference sharpened, but stability drops |
| | 8 | GRPO | $\mathbf{0.53} \pm 0.10$ | 100 | |
| | | **GRPO+DPO** | $0.48 \pm 0.25$ | 100 | Drop in GMRR |
| | 16 | GRPO | $\mathbf{0.51} \pm 0.10$ | **100** | |
| | | **GRPO+DPO** | $0.49 \pm 0.32$ | 74 | Drop in both metrics |
| | 32 | GRPO | $0.48 \pm 0.14$ | 96 | |
| | | **GRPO+DPO** | $\mathbf{0.56} \pm 0.18$ | **97** | Improvement in both metrics |
| Qwen2.5-1.5B | 2 | GRPO | $\mathbf{0.50} \pm 0.08$ | 100 | |
| | | **GRPO+DPO** | $0.27 \pm 0.24$ | 100 | GMRR Drop |
| | 4 | GRPO | $0.51 \pm 0.09$ | **100** | |
| | | **GRPO+DPO** | $\mathbf{0.56} \pm 0.18$ | 78 | Improvement but stability degradation hurts GMRR |
| | 8 | GRPO | $\mathbf{0.51} \pm 0.09$ | 100 | |
| | | **GRPO+DPO** | $0.40 \pm 0.17$ | 100 | GMRR Drop |
| | 16 | GRPO | $0.52 \pm 0.09$ | 100 | |
| | | **GRPO+DPO** | $\mathbf{0.60} \pm 0.17$ | 100 | **Strongest Improvement (GMRR & Rate)** |
| | 32 | GRPO | $\mathbf{0.52} \pm 0.09$ | 100 | |
| | | **GRPO+DPO** | $0.48 \pm 0.13$ | 100 | GMRR Drop |

*Table 4.* Performance summary of Qwen2.5-1.5B on the OOD Test Set. The **SFT (Peak)** baseline performs the worst, indicating overfitting to training optima. **GRPO** improves generalization via online exploration, while the **hybrid GRPO+DPO** pipeline achieves the highest reward by combining structural stability with preference sharpening.

| Model | Approach | GMRR | Parse Rate |
|---|---|---|---|
| Qwen2.5-1.5B | SFT (Peak) | $0.45 \pm 0.12$ | 100% |
| Qwen2.5-1.5B | ICL | $0.51 \pm 0.09$ | 100% |
| Qwen2.5-1.5B | GRPO | $0.53 \pm 0.09$ | 100% |
| Qwen2.5-1.5B | Hybrid | $\mathbf{0.60} \pm 0.17$ | 100% |

This design choice was deliberate to avoid *priming bias*, ensuring that users did not subconsciously tailor their requests to perceived system capabilities (e.g., only asking for "more bass"). As expected, this yielded highly Out-Of-Distribution requests (e.g., volume control, removing instruments, or changing speaker identity). After filtering out these functional system requests, the remaining challenging prompts were retained.

In the second phase, 11 of the original participants returned to evaluate the models' responses. For each prompt, participants listened to audio processed by both the hybrid RL model and the GPT-4o mini baseline in a randomized, blinded interface. They rated how well the resulting audio satisfied the original request on a 5-point Likert scale (1="Very Poorly" to 5="Very Well").

### 6.2. Quantitative Results

The aggregate results (Table 5) indicate that our specialized RL model performs at parity with, and on average exceeds, the previously-best generalist baseline. More specifically:

- **Mean Preference Score:** The 1.5B RL model achieved a mean rating of **3.04**, outperforming the GPT-4o mini baseline (2.92). This suggests that the RL

*Table 5.* Subjective Evaluation Results. Comparison of the proposed hybrid RL approach (1.5B) against the baseline (GPT-4o mini). Mean Likert score ($\pm$ Standard Error). The $p$-value is calculated via the Wilcoxon Signed-Rank test

| Method | Score (1–5) | Win Rate | *p*-val |
|---|---|---|---|
| GPT-4o mini (ICL) | 2.92 $\pm0.06$ | 72.1% | 0.073 |
| **Qwen2.5-1.5B (RL)** | **3.04** $\pm0.06$ | **77.9%** | |

model handles ambiguous, OOD prompts gracefully.

- **Win/Tie Rate:** In head-to-head comparisons, the RL model was preferred or rated as equivalent to the baseline in **77.9%** of trials.

- **Statistical Analysis:** We employed the Wilcoxon Signed-Rank test, which handles ordinal Likert data and paired measurements robustly for small sample sizes (Wilcoxon, 1992). The test yielded $p = 0.073$, indicating that the observed difference in ratings was not statistically significant at the conventional $\alpha = 0.05$ threshold. This suggests that the two systems exhibit comparable perceptual performance within the limits of our evaluation.

The critical implication of these results is efficiency: The proposed Density-Based RL pipeline successfully distilled the necessary audio control capabilities into a model that is orders of magnitude smaller than GPT-4o mini, achieving equivalent user satisfaction without the need for large inference compute or proprietary APIs.

## 7. Discussion and Limitations

Our results suggest a paradigm shift for LLM alignment in continuous control: Rather than training unstable proxy reward models, we can leverage the "wisdom of the crowd" directly via non-parametric density maps. However, this approach entails specific constraints that merit discussion.

### 7.1. The Curse of Dimensionality

Our Reflective-KDE approach is highly effective for low-dimensional spaces (e.g., the 2D Beosonic interface). However, KDE suffers from the curse of dimensionality (Bellman, 1957). As the control space expands (e.g., additional acoustic parameters or a 6-DoF robotic arm), the data required to populate a dense reward map grows exponentially. Scaling this framework to high-dimensional action spaces will likely require latent variable models (e.g., Variational Auto Encoders (Kingma & Welling, 2022)) to compress the control space into a tractable lower-dimensional manifold before applying our density-based RL.

### 7.2. Text-Only Conditioning

The current system aligns audio parameters solely based on the semantic intent of the text prompt (e.g., "Make it warm"). It does not analyze the input audio signal ($P(\text{Selected}|x, \text{Audio}_{ID})$). While this enables a lightweight, sparse (respond when requested), privacy-preserving inference (no audio upload required), it limits the system's context awareness. A "warm" adjustment for a podcast could differ significantly from a "warm" adjustment for techno music. Future iterations could integrate audio embeddings (e.g., CLAP (Elizalde et al., 2023) features) into the policy to enable content-adaptive equalization.

### 7.3. Consensus vs. Personalization

Finally, while our density map method effectively models the population's preference space, it does not yet solve the problem of individual personalization. By optimizing for the peaks of the density map, the hybrid model acts as a "Mode-Seeking" agent–it generates the setting that could satisfy *any* user. It does not yet have the capacity to adapt to user-specific hearing profiles. Extending this framework to conditioned density estimation ($P(\text{Selected}|x, \text{User}_{ID})$) remains a promising and natural avenue for future research.

## 8. Conclusion

In this work, we presented a framework for aligning Large Language Models to continuous, subjective control spaces without relying on unstable proxy reward models. By leveraging large-scale population data ($N \approx 90,000$), we constructed non-parametric preference density maps via Reflective Kernel Density Estimation and Probability of Selection estimation. These maps serve as a ground-truth reward surface that captures the full diversity of user preferences.

Our comparative analysis of alignment algorithms revealed a critical structural trade-off. We demonstrated that offline DPO can struggle with format adherence in continuous spaces, as they lack the on-demand negative feedback loops necessary to enforce strict syntax constraints. Conversely, online GRPO provides robust structural grounding but can converge to mediocre safe choices.

We resolved this trade-off with a hybrid pipeline, using GRPO to establish a format-compliant policy and DPO with synthetic Peak Injection to sharpen preference alignment. This approach achieved the highest alignment and compliance (GMRR 0.60, Parse rate 100%), enabling a 1.5B parameter model to achieve perceptual parity with a carefully prompt engineered GPT-4o mini model, in a blind listening test.

## Impact Statement

This paper presents work whose goal is to advance the field of Machine Learning. There are many potential societal consequences of our work, none which we feel must be specifically highlighted here.

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
