# OpenReview forum: "Direct Preference Density Alignment for Conversational Audio Equalization"
_ICML.cc/2026/Conference — Submitted to ICML 2026_

### Official Review · Reviewer_Cu6t · 2026-03-12

**Soundness:** 2
**Presentation:** 3
**Significance:** 3
**Originality:** 3
**Overall Recommendation:** 3
**Confidence:** 3

**Summary:**

This paper proposes Direct Preference Density Alignment, a proxy-free alignment framework for continuous control. It replaces learned reward models with non-parametric preference density maps estimated from large-scale user interaction data ($N\approx90,000$) using Reflective-KDE and a ratio-of-densities approach in a bounded 2D EQ parameter space. The authors compare offline DPO and online GRPO, identifying a trade-off where DPO struggles with strict output formatting while GRPO favors safe, format-compliant outputs. They introduce a hybrid GRPO+DPO pipeline with "peak injection" to combine these strengths. Evaluated on an Out-Of-Distribution (OOD) prompt set, the 1.5B hybrid model reaches a Greedy Mean Relative Reward (GMRR) of 0.60 with a 100% parse rate. A blind listening test suggests perceptual parity with a GPT-4o mini baseline, though the results are not statistically significant ($p=0.073$).

**Compliance With Llm Reviewing Policy:**

Affirmed.

**Key Questions For Authors:**

1. **DPO Baselines & Constraints:** Did you experiment with simple decoding constraints (e.g., regex, grammar, function calling) during DPO training/inference to mitigate format collapse? How does DPO compare to GRPO under identical strict output constraints?
2. **Dataset Construction:** How exactly were the DPO baseline pairs constructed prior to the hybrid stage? Please clarify the sampling distribution, pair count per prompt, and coverage over the 2D space.
3. **Hybrid Instability:** Several hybrid configurations in Table 3 show significant degradation compared to base GRPO. Could you provide multi-seed averages with confidence intervals to clarify variance, and explain the selection protocol that led to the headline 0.60 GMRR result?
4. **Ablations:** Can you provide ablations on the KDE bandwidth selection (specifically the $0.25$ scaling factor on Scott's Rule), the $\epsilon$ term for ratio stability, and the number of injected peaks ($K$)? How sensitive are the GMRR and parse rates to these choices?
5. **Subjective Evaluation Metrics:** In the listening test, how exactly are ties defined and aggregated into the reported "win rate"? Given the small sample size ($N=11$), a power analysis would help contextualize the $p=0.073$ result.

**Limitations:**

The authors candidly discuss the curse of dimensionality restricting this to low-dimensional spaces, the lack of audio-conditioned control, and the absence of user personalization. However, the limitations section should ideally also address the reliance on purely semantic OOD generalization and the small sample size of the subjective evaluation.

**Strengths And Weaknesses:**

**Originality:**

The paper presents a novel instantiation of proxy-free alignment for continuous control. Estimating a bounded, debiased reward surface via Reflective-KDE and density ratios to avoid reward-model instability is a creative and domain-aware approach to multi-modal preference landscapes.

**Significance:**

**Strength:** The authors demonstrate that a compact, open model (1.5B parameters) can match a strong proprietary baseline (GPT-4o mini) on a practical audio control task using only logged interactions. This offers valuable insights for cost-effective deployment and continuous-control alignment.

**Weakness:** The approach is currently restricted to a very low-dimensional control space (2D). While the limitations section acknowledges the curse of dimensionality, the paper offers limited evidence or ablations for higher-dimensional feasibility.

**Soundness:**

**Strength:** The evaluation is multi-faceted, utilizing objective metrics (prompt-wise GMRR, parse rate) and a perceptual A/B test. Systematic sweeps over GRPO group size and KL penalties ($\beta$) illuminate stability and exploration trade-offs.

**Weakness:** The diagnosis of DPO's "format collapse" may conflate algorithmic limitations with engineering choices. The paper does not compare against constrained DPO variants (e.g., C-DPO) or simple grammar-constrained decoding, which could directly enforce format validity during DPO.

**Weakness:** Objective results for the hybrid model show instability across configurations, with several runs in Table 3 degrading substantially. Furthermore, the A/B listening test is quite small ($N=11$ participants) , and the resulting $p=0.073$ does not cross standard significance thresholds, meaning claims of "parity" must be framed carefully.

**Presentation:**

**Strength:** The motivation for proxy-free alignment in subjective spaces is well-articulated, and the reward surface construction is explained clearly with intuitive figures.

**Weakness:** Reproducibility details are sparse. Critical training hyperparameters, decoding settings, explicitly defined validity checkers, and pair construction details for the DPO baseline are missing or under-specified. Additionally, some tables contain formatting artifacts that reduce readability.

---

> ### Author Rebuttal · Authors · 2026-03-30
>
> We thank the reviewer for recognizing our approach's originality and rigorous evaluation. We address your constructive feedback below.
>
> **Decoding constraints:** We intentionally omitted inference-time constraints to investigate the base algorithms' fundamental dynamics. While these methods guarantee valid formatting, they mask the model's true learned distribution. Applying them would obscure whether vanilla DPO inherently learns structural constraints as effectively as online GRPO. We agree, however, that constrained decoding is highly valuable for practical deployment.
>
> **DPO dataset construction:** The DPO baseline dataset follows the exact two-step protocol from the "Synthetic Preference Mining and Peak Injection" stage, but uses the original LLM policy as the reference instead of the post-GRPO policy. As outlined in the text, we perform on-policy mining by sampling until we obtain K positive (high reward) and K negative (low reward) completions, and if the model fails to find enough peaks, we analytically inject the local maxima/minima to the remaining spots in the pools. By combining the positive and the negative completions, the resulting pair count per prompt is $K^2$. In our study we set K = 10. We will emphasize the parallel between the standard DPO baseline and the hybrid stage more prominently in the revised text.
>
> **Hybrid instability and multi-seed averages:** Hybrid instability likely stems from using a universal DPO regularization ($\beta$). We agree multi-seed confidence intervals improve soundness and will include this sensitivity analysis. For checkpoint selection, we chose the model maximizing validation reward and reported OOD test set results. These details will be explicitly stated.
>
> **Ablations (KDE bandwidth, $\epsilon$, and K):**
> The 0.25 scaling factor on Scott's Rule accounts for two specific properties of our space: a 0.5 factor to adjust for the extra point mass introduced by the Reflective boundary, and an additional 0.5 factor to counteract the known tendency of Scott's Rule to over-smooth highly non-convex spaces. Because the bandwidth inherently defines the ground-truth reward surface, standard ablations (which require a fixed test set) are ill-posed. However, we appreciate the insightful request, and we will include visual ablations of the density maps in the revision to demonstrate how this scale captures the underlying data distribution without over-smoothing. The stability term used in the density ratio calculation is an epsilon constant set to 1e-7. Because this epsilon functions as a numerical safeguard to prevent division by zero, varying it (e.g., to 1e-6 or 1e-8) does not meaningfully alter the density computations or the resulting preference gradients. While an ablation would not yield measurable differences, we completely agree it is essential for transparency and will explicitly document the 1e-7 value in the revision. The maximum choice of K is bounded by the actual number of local maxima present in the density map (pushing K forces the injection of lower reward peaks). We completely agree that analyzing the model's sensitivity to fewer peaks is valuable. In the revised manuscript, we will include an ablation study comparing our baseline (K=10) with a reduced number of injected peaks (e.g., K=5) to explicitly demonstrate how this density of "Golden Truths" impacts the resulting GMRR and parse rates.
>
> **Subjective evaluation metrics and sample size:**
> We appreciate the rigorous check on our statistical reporting. While the user cohort was $N=11$, each participant evaluated 30 distinct Out-Of-Distribution prompts, resulting in 330 total pairwise evaluations. We realized that we never clarified this in the original manuscript and we apologize. We completely agree with the reviewer that $p=0.073$ does not cross the $\alpha=0.05$ threshold for strict statistical superiority. Because of this, we were careful to claim only perceptual parity in the manuscript. Given that our 1.5B model achieved a higher mean score (3.04 vs 2.92) and was preferred or rated equivalently in 77.9\% of the 330 trials against a vastly larger proprietary model, we feel that "parity" accurately and conservatively reflects the system's performance. This parity is particularly notable because our RL-tuned model achieved it purely by learning from the non-parametric reward signal. In contrast, the GPT-4o mini baseline relied on In-Context Learning with a sophisticated system prompt containing explicit semantic-to-coordinate expert mappings. Reaching comparable performance without relying on domain heuristics highlights the strength of our agnostic alignment approach, particularly for domains where such expert knowledge is unavailable. We appreciate the valuable suggestion, and to provide further transparency, we will include a post-hoc power analysis in the revised appendix.
>
> We hope these commitments fully address your concerns and respectfully ask you to consider raising your score.

---

> > ### Author Rebuttal · Reviewer_Cu6t · 2026-04-05
> >
> > I appreciate the authors' response to my concerns. While the rebuttal provides necessary clarifications, my overall assessment remains a 3 for the following reasons:
> >
> > **Statistical Significance**: The authors clarified the trial count, but the fact remains that the results ($p=0.073$) do not reach the standard statistical significance threshold. This makes the claim of "parity" with GPT-4o mini less conclusive than presented. **Hybrid Instability**: Table 3 still shows significant performance drops in several hybrid configurations. While the authors promise to add multi-seed averages, the current data suggests the method's sensitivity to hyperparameters ($\beta$) is a notable weakness. **Clarification of Details**: I acknowledge the technical details provided for $\epsilon$ and the KDE bandwidth scaling. These improve the paper’s transparency but do not fundamentally change the observed performance trade-offs.
> >
> > In summary, the paper offers an interesting direction for proxy-free alignment in continuous control, but due to the marginal statistical results and training instability, I am not inclined to raise my score.

---

> > > ### Author Response · Authors · 2026-04-07
> > >
> > > We thank the reviewer for their insight and continued engagement. We completely respect your assessment, but perhaps there is still some merit in discussing our claim of parity for the sake of clarity. In our Wilcoxon test, the null hypothesis is that the models perform equally. A result of $p=0.073$ means we fail to reject the null hypothesis; i.e., we failed to find a statistically significant difference between our 1.5B model and the GPT-4o mini baseline. Failing to establish a difference, combined with our model's higher mean score, is what supports our claim of parity. We thank you again for your time and for helping us hold our work to a higher standard!

---

### Official Review · Reviewer_JkTZ · 2026-03-13

**Soundness:** 3
**Presentation:** 3
**Significance:** 2
**Originality:** 2
**Overall Recommendation:** 2
**Confidence:** 3

**Summary:**

This paper presents a framework for aligning LLMs with subjective human preferences in continuous control tasks. The Authors specifically focus on conversational audio equalization. To avoid the instability and reward hacking associated with learned proxy reward models, the authors propose constructing non-parametric preference density maps using Reflective-KDE from a large-scale user dataset. The investigation reveals a stability gap where offline DPO fails to maintain strict output formatting in continuous spaces, leading them to propose a hybrid GRPO+DPO pipeline that combines the best of both worlds: structural robustness of online group optimization with the precision of offline refinement.

**Compliance With Llm Reviewing Policy:**

Affirmed.

**Final Justification:**

I appreciate the effort on the Authors' part during the rebuttal, however missing sections of the work yield an overall incomplete submission. I look forward to seeing a complete and updated revision of this work in the future, and will maintain my score.

**Key Questions For Authors:**

- How do you plan to address the curse of dimensionality when moving beyond a 2D control space? Are there specific latent variable models/VAEs you have already tested to compress high-dimensional preference manifolds? This is a critical question, since this space is dominated by extremely high-dimensional problems.

- Did you experiment with adding a validity penalty or a constrained output space directly into the DPO loss to see if it could match GRPO's structural stability?

- What were the parameters of the Blind A/B test? Are the Appendices intentionally missing?

**Limitations:**

Yes.

**Strengths And Weaknesses:**

**Strengths**

- The use of Reflective-KDE to transform raw population data into a ground-truth reward surface is effective and eliminates the need for unstable learned reward models or value networks.

- The GRPO+DPO method addresses the format collapse common in continuous control by using online exploration to establish structural grounding before applying targeted preference refinement.

- The framework demonstrates significant model distillation. Hence a small 1.5B-parameter model achieves subjective performance comparable to GPT-4o mini, which reduces inference compute requirements.

**Weaknesses**

- The approach is currently limited to low-dimensional control spaces (2D). Scaling to high-dimensional tasks like complex robotics poses as a challenge due to the curse of dimensionality. The Authors acknowledge this in section 7.1.

- The model aligns parameters based on text prompts without analyzing the input audio signal. Therefore it cannot provide content-aware adjustments. More discussion on how this limits the context awareness will be helpful.

- The method seeks the peaks of a population-wide density map, and the model acts as a mode-seeking agent instead of adapting to the hearing profiles or preference tastes of individual users. Therefore the method, although presenting points of novelty, cannot be considered a paradigm shift in LLM alignment (as stated in the paper).

---

> ### Author Rebuttal · Authors · 2026-03-30
>
> We sincerely thank the reviewer for rating our Soundness and Presentation highly, and for recognizing that our Reflective-KDE approach effectively eliminates the need for unstable proxy reward models. We appreciate your constructive feedback and address your specific questions below.
>
> **Regarding the curse of dimensionality and latent variable models:**
> We completely agree with your assessment in Section 7.1 regarding the challenge of scaling to high-dimensional spaces. While we have not yet RL-tuned with VAEs for an alternative task, the broader field of Neural Audio Codecs has already proven that highly complex, continuous audio spaces can be robustly compressed into low-dimensional latent manifolds that can be effectively processed by LLMs. Integrating these established compression techniques with our density-based RL framework is a promising and feasible direction for future work, making our foundational 2D proof-of-concept highly extensible.
>
> **Regarding text-only conditioning and context awareness:**
> We appreciate the reviewer highlighting the absence of direct audio signal processing. We wish to clarify that our core contribution is the alignment pipeline itself, which is entirely agnostic to the input modality. The lack of an audio encoder does not affect the validity of the RL framework; the exact same Reflective-KDE and GRPO+DPO methodology can seamlessly incorporate audio embeddings (e.g., via a multimodal LLM) to enable content-aware adjustments in the future. We will add a dedicated discussion in the revision detailing how audio context can be integrated into this pipeline.
>
> **Regarding the mode-seeking behavior and personalization:**
> The reviewer correctly notes that the model acts as a mode-seeking agent. Because conversational audio equalization is highly subjective, no single global solution exists. In tasks that are inherently unsolvable at a universal level, establishing a model that captures population-level tendencies is a good starting point. It acts as a foundational prior that bypasses the "cold-start" problem for any future personalized recommender system. Furthermore, our claim of a "paradigm shift" refers primarily to the methodology: moving away from unstable proxy reward models toward non-parametric, data-driven density maps.
>
> **Regarding explicitly constrained output spaces (e.g., C-DPO):**
> While incorporating explicit validity penalties like C-DPO is a highly valuable direction, we deliberately evaluated vanilla DPO and GRPO to cleanly isolate the fundamental "stability gap" without introducing variables from custom penalties. We agree this is important context and will expand our Related Work section to acknowledge explicitly constrained DPO variants.
>
> **Regarding the missing Appendix and A/B test parameters:**
> We sincerely apologize for the omission of the Appendix and will ensure it is included in the revision. To immediately address the reproducibility of the blind A/B test (Section 6.1), the parameters were as follows: 11 participants evaluated audio on a randomized, blinded web interface using standardized volume levels on the same pair of headphones. Participants evaluated both models on 30 Out-Of-Distribution prompts. We will ensure these operational details are fully documented in the Appendix that will be included in the revised manuscript.
>
> We hope these clarifications and the provided evaluation details resolve your concerns, and we respectfully ask that you consider raising your score in light of these revisions.

---

> > ### Author Rebuttal · Reviewer_JkTZ · 2026-04-04
> >
> > Thank you for the rebuttal, I appreciate the effort on the Authors' part to address my concerns on a high level. I believe the work has merit and can be improved, for example by including a discussion on high-dimensional tasks.
> >
> > However, the missing Appendix and supporting documentation is difficult to visualize at this point. Although my concerns and questions were mildly discussed, I do not many actionable items that are telling of what will be included in the Revision. Therefore I view this as an incomplete submission in its current state, and encourage the Authors to further complete this work. Thank you.

---

> > > ### Author Response · Authors · 2026-04-07
> > >
> > > We thank the reviewer for their time. We completely understand the position regarding the missing appendix. Omitting it was an unfortunate error on our part, and we agree it limits the immediate reproducibility of the submission. We appreciate the constructive feedback provided regardless, which will strengthen our next revision.

---

### Official Review · Reviewer_r8xu · 2026-03-14

**Soundness:** 3
**Presentation:** 2
**Significance:** 2
**Originality:** 2
**Overall Recommendation:** 3
**Confidence:** 3

**Summary:**

The paper introduces a novel framework for aligning LLMs to subjective, continuous-control tasks, specifically conversational audio equalization, without relying on unstable proxy reward models. The authors construct a ground-truth preference density map using Reflective Kernel Density Estimation on a large-scale user dataset. Through empirical analysis, they demonstrate that standard offline DPO struggles with format collapse in continuous spaces, whereas online Group Relative Policy Optimization (GRPO) maintains strict structural adherence but often yields sub-optimal predictions. To resolve this trade-off, the paper proposes a hybrid GRPO+DPO. The resulting 1.5B-parameter model achieves the performance improvement and perceptual parity with a heavily prompt-engineered GPT-4o mini baseline in a blind A/B listening test.

**Compliance With Llm Reviewing Policy:**

Affirmed.

**Final Justification:**

The rebuttal has addressed the initial concerns, but the overall novelty of the work seems limited. I maintain my initial score.

**Key Questions For Authors:**

See weaknesses above.

**Limitations:**

yes

**Strengths And Weaknesses:**

**Strengths**

* The use of Reflective Kernel Density Estimation to build a non-parametric reward surface directly from population data is a clever approach. This bypasses the instability and reward hacking vulnerabilities commonly associated with learned proxy reward models in continuous control.

* The objective metrics are robustly backed by a blind A/B subjective listening test. This real-world validation demonstrates that the 1.5B parameter hybrid model achieves perceptual parity with a heavily prompt-engineered GPT-4o mini baseline.

**Weaknesses**

* In Section 3.5, how was the aggregation of synonymous clusters performed?

* While the formulation of the non-parametric reward surface via Reflective-KDE is an interesting and valid approach to bypassing proxy reward models, the overall algorithmic novelty of the training framework is somewhat limited. The paper relies heavily on standard, off-the-shelf implementations of DPO and GRPO. The proposed hybrid pipeline, while effective for this specific task, is primarily a sequential engineering application of existing methods.

---

> ### Author Rebuttal · Authors · 2026-03-30
>
> We sincerely thank the reviewer for highlighting the merits of our work, specifically noting that our KDE approach is clever and our objective metrics are robust. We have addressed your concerns below.
>
> **Regarding the aggregation of synonymous clusters:**
> We thank the reviewer for this question. We initially attempted to automate the synonym-sentence generation process using LLMs. However, we observed that LLMs tended to overcomplicate the task, adding unnecessary complexity to simple intents (e.g., "Increase the warmth" becoming "Make it warmer" or "Add more warmth"). To ensure a clean, high-quality dataset, we found it more effective to manually curate the synonymous clusters. We then evaluated the similarity of these sentences using Sentence-BERT to verify the groupings and catch any potential human errors. We will add this explanation to Section 3.5 in the revised version of the paper to ensure our density estimation is built on a precise and validated foundation.
>
> **Regarding algorithmic novelty and sequential engineering:**
> We appreciate the reviewer's perspective, and we wish to clarify that we see our framework not merely as sequential engineering, but as a novel paradigm for preference alignment. Consider the current landscape: PPO is very effective but requires memory-intensive Reward and Value models. GRPO removes the value model, but advantage estimation is sub-optimal. DPO removes both models, but loses the benefits of online exploration by shifting to offline learning.
>
> To our knowledge, our approach is the first to remove both Value and Reward models whilst strictly maintaining the online RL nature of the optimization. By using Reflective-KDE to build a reward surface from offline data, we enable true online exploration (via GRPO) without the GPU overhead or the "reward-hacking" risks associated with learned proxy models. We believe bridging this gap between offline efficiency and online exploration is a significant conceptual contribution to the field.
>
> We hope that our arguments satisfy your concerns regarding our manual cluster curation and the conceptual novelty of enabling proxy-free online RL. We respectfully ask that you consider raising your score in light of these updates.

---

> > ### Author Rebuttal · Reviewer_r8xu · 2026-04-06
> >
> > I appreciate the authors' detailed response. However, I still think the proposed algorithmic novelty appears limited for an ICML conference.

---

> > > ### Author Response · Authors · 2026-04-07
> > >
> > > We thank the reviewer for engaging with our rebuttal. We respectfully acknowledge your assessment regarding the algorithmic novelty expected at ICML. We greatly appreciate your positive remarks regarding the cleverness of our KDE approach and the robustness of our metrics. Thank you again for your time and for helping us refine this work.

---

### Official Review · Reviewer_eaHn · 2026-03-14

**Soundness:** 3
**Presentation:** 3
**Significance:** 2
**Originality:** 2
**Overall Recommendation:** 3
**Confidence:** 3

**Summary:**

Large Language Model alignment methods typically rely on learned proxy reward models. This paper proposes an alternative that removes the proxy, aligning models through preference density maps constructed directly from large-scale user data, and this approach is instantiated on a continuous audio equalization control task. This paper demonstrates that standard offline DPO, in contrast to online GRPO, struggles to adhere to pre-defined output formats. Building on this insight, a hybrid GRPO+DPO training pipeline is proposed. This framework can achieve the best performance across objective metrics and enables a Qwen 2.5 1.5B model to achieve comparable performance to a carefully prompt-engineered GPT-4o mini baseline.

**Compliance With Llm Reviewing Policy:**

Affirmed.

**Final Justification:**

Thank the authors for the detailed response. Regarding the algorithmic novelty, the response clarifies this work's major contribution. However, I think the algorithmic contribution remains limited, since most components are well established in prior work. And this work's novelty mainly lies in well combining these components.

Regarding the high variance, the rebuttal attempts to view it as an advantage for a subsequent personalization phase. However, in my understanding, this is not the main claim of the current paper. While this may partially explain the issue, it does not fully resolve the main concern in the current version.

Overall, considering that the response resolves part of my concerns, I am willing to raise my score to 3. But the major issues still remain.

**Key Questions For Authors:**

Please see the section of Strengths And Weaknesses.

**Limitations:**

Yes

**Strengths And Weaknesses:**

**Advantage:**

This paper is well structured. The proposed method bypasses the instability of learned Reward Models. The paper identifies specific failure modes of offline DPO in continuous control and demonstrates the structural superiority of online GRPO, which is insightful. The experimental results are impressive. The proposed GRPO+DPO framework achieves the best performance across various metrics and enables a 1.5B model to achieve comparable performance to GPT-4o mini.

**Weakness:**

- The discussion of related works for GRPO is limited in Section 2.3. There have been a number of GRPO variants. The authors need to expand this part to include more variants of GRPO for further analysis.

- The algorithmic novelty appears limited. The proposed method in this paper mainly builds on the existing methods, such as Reflective-KDE, DPO, and GRPO. The hybrid GRPO+DPO training pipeline seems incremental.

- In Table 3, GRPO and GRPO+DPO reach their peak performance at different numbers of rollouts, which exhibits a potential mismatch on this hyperparameter. Moreover, the GRPO+DPO performance (in terms of GMRR and Parse Rate) fluctuates as the number of rollouts increases. It is different from GRPO, whose performance first increases and then slightly declines or remains unchanged. This raises a concern about the stability of the proposed GRPO+DPO approach. If the method is indeed sensitive to the number of rollouts, the paper needs to provide a clearer suggestion on how the number of rollouts can be set in practice.

- For the proposed GRPO+DPO method, the results exhibit higher variance. It would be helpful to discuss why variance increases and whether this affects practical reliability.

---

> ### Author Rebuttal · Authors · 2026-03-30
>
> We sincerely thank the reviewer for their careful evaluation and for recognizing that our framework is well-structured and our experimental results are impressive. We appreciate the constructive feedback and have addressed your concerns below.
>
> **Regarding the discussion of GRPO variants:**
> We thank the reviewer for this valuable suggestion. During our preliminary experiments, we implemented and tested recent GRPO variants, including DAPO and Dr.GRPO. However, we found that for our specific continuous control task, these variants did not provide a tangible performance advantage over the base algorithm. To maintain a clean, focused analysis of the fundamental structural trade-offs between online (GRPO) and offline (DPO) alignment, we deliberately chose to evaluate the original algorithms. We completely agree that discussing these variants adds valuable context, and we will expand Section 2.3 in the revision to cite and discuss these works.
>
> **Regarding algorithmic novelty:**
> We wish to clarify our conceptual contribution. While the underlying components (KDE, GRPO, DPO) are indeed established, our framework represents a fundamental shift in how preference alignment can be achieved in continuous spaces. Consider the current landscape of alignment algorithms: PPO offers peak performance but requires maintaining separate Reward and Value networks in memory. GRPO improves this by removing the Value model. DPO removes both the Value and Reward models, but critically, it shifts the optimization to offline learning, which we demonstrate compromises structural adherence in continuous control.
>
> To our knowledge, our approach is the first to remove both the Value and Reward models while strictly maintaining the online RL nature of the optimization. By leveraging Reflective-KDE to construct a non-parametric reward surface from offline preference data, we enable the model to perform true online exploration (via GRPO) without the massive GPU overhead or reward-hacking risks associated with a learned proxy reward model. We believe that bridging this gap between offline preference data and online proxy-free RL is a highly impactful conceptual contribution.
>
> **Regarding hyperparameter mismatch and stability (Rollouts / $\beta$):**
> We thank the reviewer for this sharp observation. We believe the fluctuation observed in the GRPO+DPO pipeline stems primarily from the DPO phase's sensitivity to the reference model. Generally, increasing the number of rollouts provides better context for advantage estimation in the initial GRPO phase. This means that GRPO models trained with different rollout counts converge with slightly different baseline characteristics.
>
> The instability in the hybrid approach arises because we applied a universally fixed $\beta$ value (KL-penalty) for the DPO phase across all of these varying reference models. To address your question regarding practical application: the most effective way to control this instability is to tune the $\beta$ parameter during the DPO phase, tailoring the regularization to the specific GRPO reference model used rather than relying on a fixed default.
>
> **Regarding variance and practical reliability:**
> The increased variance in the GRPO+DPO method reflects a classic exploration-exploitation dynamic. Because the model is learning from a non-parametric density map rather than a deterministic proxy, the higher variance indicates that the model is capturing a broader area of the optimal preference manifold. In highly subjective domains like audio control, this variance could actually be advantageous; as we discuss in Section 7.3, it acts as a robust, population-level prior that is well suited for a subsequent individual personalization phase.
>
> We hope these clarifications resolve your concerns regarding our conceptual novelty and hyperparameter dynamics, and we hope you will consider raising your score in light of these revisions.

---

> > ### Author Rebuttal · Reviewer_eaHn · 2026-04-03
> >
> > Thank the authors for the detailed response. Regarding the algorithmic novelty, the response clarifies this work's major contribution. However, I think the algorithmic contribution remains limited, since most components are well established in prior work. And this work's novelty mainly lies in well combining these components.
> >
> > Regarding the high variance, the rebuttal attempts to view it as an advantage for a subsequent personalization phase. However, in my understanding, this is not the main claim of the current paper. While this may partially explain the issue, it does not fully resolve the main concern in the current version.
> >
> > Overall, considering that the response resolves part of my concerns, I am willing to raise my score to 3. But the major issues still remain.

---

> > > ### Author Response · Authors · 2026-04-07
> > >
> > > We thank the reviewer for taking the time to read our rebuttal and for the willingness to raise the score. We completely respect your perspective regarding the algorithmic novelty and your remaining concerns about variance. Your feedback has highlighted important areas where we must better justify our framework's stability and contributions, and we will be sure to incorporate these insights into our future revisions. Thank you again for your time and your constructive evaluation.

---

### Decision · Program_Chairs · 2026-04-30

**Decision:**

Reject

**Comment:**

This paper proposes a proxy-free alignment framework for continuous audio equalization control, using Reflective-KDE to construct non-parametric reward surfaces from large-scale user data and a hybrid GRPO+DPO pipeline to address DPO's format collapse in continuous spaces. The reviewers acknowledge the creative use of Reflective-KDE to bypass learned reward models, the practical value of enabling a 1.5B model to approach GPT-4o mini performance, and the inclusion of both objective metrics and a subjective A/B listening test.

However, all four reviewers independently raised concerns about limited algorithmic novelty, noting that the framework primarily combines established components (KDE, GRPO, DPO) in a sequential pipeline. Additional concerns include: the hybrid model's instability across configurations (Table 3), the A/B test not reaching statistical significance (p=0.073, N=11), the missing Appendix undermining reproducibility, and the absence of comparisons with constrained DPO variants that could directly address format collapse without the hybrid pipeline.

After rebuttal, none of these core concerns were adequately resolved—no new experimental evidence was provided, and all four reviewers maintained their original scores. The novelty limitation is fundamental and cannot be addressed through textual clarification alone, and the missing Appendix renders the submission incomplete. I therefore recommend rejection.